# Manipulating Infrared Emissivity with Galvanized Iron Sheets for Physical Adversarial Attack

## Abstract

For adversarial attacks on infrared detectors, previous works have focused on designing the physical patches through temperature variations, overlooking the impact of infrared emissivity on infrared imaging. In fact, infrared emissivity significantly affects infrared radiant intensity at the same temperature. In this paper, a QR-like adversarial attack patch is designed by manipulating the surface emissivity of objects to alter the infrared radiation intensity emitted from the object's surface, called Emissivity QR-like Patch (E-QR patch). In this paper, the surface emissivity of the object is manipulated through the adjustment of surface roughness. Various levels of surface roughness are realized by a commonly used metal material, galvanized iron sheets, to produce physically adversarial patches with diverse infrared radiation intensity. Considering the possible transformation distributions between the digital and physical domains, a physical E-QR patch, which is robust to noise, angle, and position, is generated by an expectation over the transformation framework. Smoothing loss is incorporated to minimize the loss in physical reconstruction, thereby effectively mitigating shooting errors in the physical domain induced by abrupt pixel changes in the digital domain. Experimental results show that the E-QR patch achieves more than $80\%$ attack success rate for infrared pedestrian detectors in a physical environment.

## 1 Introduction

Infrared object detector that trained by Deep Neural Networks (DNNs) has received extensive attention within the field of computer vision because of their significant performance. Infrared object detector is widely used in pedestrian detection (Biswas & Milanfar, 2017), autonomous driving (Dai et al., 2021) due to its unique advantages, such as night imaging, temperature measurement and others. (Szegedy et al., 2013)

However, DNNs' lack of interpretability and robustness makes it vulnerable to attack. Szegedy et al. (2013) first discovered that the DNNs-based image classifier is susceptible to malicious devised noise, which make DNNs output incorrect results with high confidence. These images with added adversarial perturbations do not look different from clean images to the human eye. The process described above is known as adversarial attack. Adversarial attacks can be classified into white-box attacks, black-box attacks and gray-box attacks (Akhtar & Mian, 2018; Kloukiniotis et al., 2022). In white-box attack, the attacker has total knowledge of the targeted network, the adversary can easily detect potential vulnerabilities of the targeted model and generate strong attacks fooling easily the model (Goodfellow et al., 2014a; Moosavi-Dezfooli et al., 2016a; Madry et al., 2017; Jiang et al., 2022). Whereas, in black-box attack, the structure of the targeted architecture and its parameters are unknown to the adversary. the adversary can observe the model outputs to receive some substantial properties and compromise the targeted model (Li et al., 2022; Su et al., 2019a; Liu et al., 2016). Attacks between white box and black box attacks are called gray-box attacks. The inducibility of attack can be categorized into target and non-target attacks (Akhtar & Mian, 2018). In non-targeted attacks, the adversary only needs to add perturbations to make the target model produce wrong results, while a targeted attack needs to make the DNNs model produce wrong results toward an intended target. Some works show that adversarial examples can exist not only in the digital world

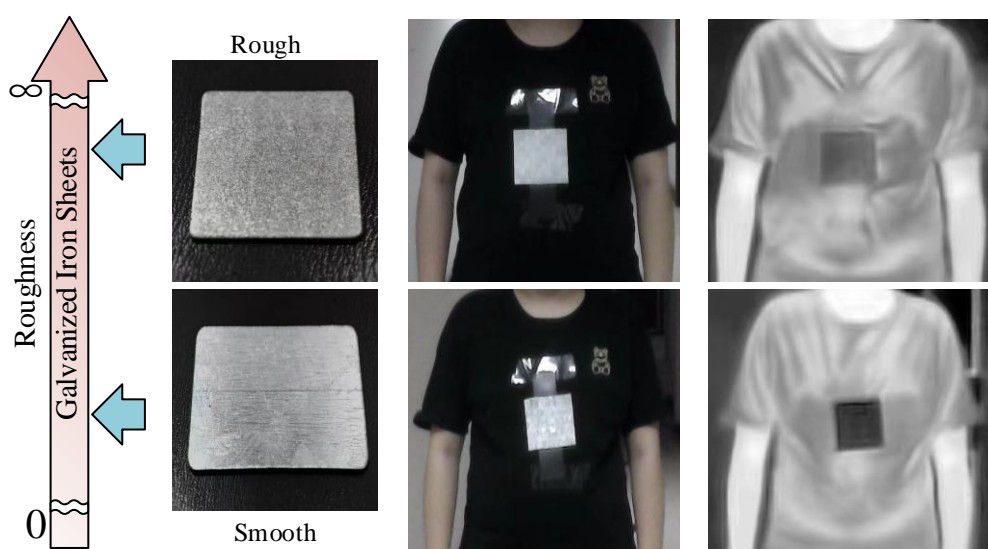

Figure 1: Relationship between roughness and infrared pixels. This figure shows the galvanized iron sheets with different degrees of roughness, which results in different pixel values captured by the same infrared sensor.

Table 1: Infrared physical adversarial attack methods.

| Methods | Descriptions | Materials | Physical factors |
|---|---|---|---|
| Zhu et al. (2021) | Multiple Small Bulbs on a Cardboard. | Tungsten | Temperature |
| Bendelac et al. (2021) | Heat Generating Resistors are Sorted in an Array. | Resistor | Temperature |
| Zhu et al. (2022) | A Clothe with QR Code Made by Aerogel | Aerogel | Temperature |
| Wei et al. (2023a) | Pure Color Blocks with Position and Shape. | Warming and Cooling Paste | Temperature |
| Hu et al. (2023) | Blocks with Position, Angle, Length, and Color | Warming and Cooling Paste | Temperature |
| Wei et al. (2023b) | Aerogel with Irregular Patterns Patch. | Aerogel | Temperature |
| Ours Method | Different Roughness of Sheets. | Galvanized Iron Sheets | Emissivity |

(Kurakin et al., 2016; Moosavi-Dezfooli et al., 2016b; Su et al., 2019b), but can also pose a threat to detectors in the physical world (Athalye et al., 2018; Duan et al., 2021; Wu et al., 2020).

Recently, there have been studies on infrared adversarial attacks, utilizing the infrared properties of different materials. To our knowledge, adversarial examples based on light bulbs (Zhu et al., 2021) and resistors (Bendelac et al., 2021) are implemented through electrical control energy-consuming materials. However, it requires continuous power consumption and imposes significant restrictions on the physical placement of patches on the object. In order to solve the non-portability and stealthiness of electronically controlled patches, (Wei et al., 2023a; Hu et al., 2023) employed the chemical reaction of the warming and cooling pastes to attack the infrared detector. However, it is difficult for the above chemical reaction to control the surface temperature precisely for a long period. Then some works utilized the insulation materials to alter the the object's surface temperature. The adversarial attacking methods based on insulation materials (Zhu et al., 2022; Wei et al., 2023b) used aerogel's thermal insulation properties to change the object's surface temperature, thus creating a binary adversarial patch. Overall, the existing infrared adversarial attack methods rely on manipulating the surface temperature of objects, resulting in a limited richness of textures in infrared images.

To address the aforementioned challenges, we introduce another physical factor affecting infrared radiation intensity: infrared emissivity. Infrared imaging is influenced by temperature and infrared emissivity on infrared radiation intensity (Hou et al., 2022). By using materials with different infrared emissivity, different infrared radiation intensities can be generated at the same temperature without deliberately changing the surface temperature of the object. Diversified emissivity can be

achieved not only by using a variety of material properties but also by using different roughness of a material (Zhang et al., 2023). We introduced a common substance, a galvanized iron sheet, and polished it into different roughness as raw materials for physical adversarial patches. As depicted in Figure 1, two galvanized iron sheets exhibit completely different pixel values captured by the same infrared sensor. It is obvious that adversarial patches with intricate texture structures can be achieved by using galvanized iron sheets with different degrees of roughness.

In this paper, we propose a physically easy-to-implement infrared attack method called "Emissivity QR-like Patch" (E-QR patch). First, the galvanized iron sheets with different degrees of roughness are used to control the emissivity of the object's surface. Specially, these galvanized iron sheets are reshaped as a QR-like patch absorbed on a soft magnetic sheet though magnetic force. Second, the smoothing loss is incorporated to minimize the the loss in physical reconstruction. Finally, the position and the degree of roughness of the galvanized iron sheets are considered as the decision variables, which are determined by the black-box optimization. It is obvious that the proposed E-QR patch is easy to implement. Since these galvanized iron sheets are fixed on the soft magnetic sheet through magnetic force, the resulting E-QR patch is a reusable physical adversarial carrier. Our main contributions are summarized below:

- Infrared emissivity is considered as a new physical factor for adversarial attack and galvanized iron sheets with different degrees of roughness are used to control the emissivity of the object's surface.

- An E-QR patch is designed by searching the position and the degree of roughness of the galvanized iron sheets. The resulting E-QR patch is reusable since the introduction of the sort magnetic sheet.

- The experiments on both digital and physical domains demonstrate the effectiveness of the proposed method for attacking infrared detectors.

The remainder of the paper is organized as follows: Section II provides a brief overview of adversarial attacks and digital-physical modeling. Section III describes the E-QR patch method presented in this work. Section IV presents the experimental findings. Finally, Section V concludes the paper and looks ahead.

## 2 RELATED WORK

In this section, the fundamental idea of infrared physical adversarial attacks is first introduced. Second, the background of digital-physical modeling is discussed. Furthermore, the paper's research motivation is explained.

### 2.1 INFRARED PHYSICAL ADVERSARIAL ATTACKS

The current adversarial attacks against infrared images are shown in Table 1. Zhu et al. (2021) used the heat from small light bulbs to build an infrared adversarial patch on a circuit board to attack the infrared pedestrian detector. But the small bulb board can only attack at a specific angle of human body and not stealthy and easy to implement. In order to solve this problem, they design a "wearable" attack (Zhu et al., 2022) by a piece of clothes. It uses aerogel to make a QR-like adversarial texture and change the adversarial examples from 2D into 3D as a clothe to insulate thermal, which can change the pixel values of the pedestrian surface. Although the above approach achieves an effective attack on the physical world, the adversarial medium is attention-grabbing and look unnatural to human. To realize a physically stealthy and easy to implement infrared attack, a method (Wei et al., 2023a) achieved by putting the cooling and warming paste inside clothes to change temperature as a adversarial patch while another work (Wei et al., 2023b) reshapes the type of aerogel patch into a irregular shape.

Although the existing methods have made good progress, they are limited by temperature-controlled mode, which makes it challenging to achieve the effect of low-cost and prolonged attacks. Most of them can only define the patch as a binary pattern, reducing the attack success rate.

## 2.2 DIGITAL-PHYSICAL MODELING

Physical adversarial attacks need to face the complexity of real-world environments, which requires strong robustness of the generated adversarial examples. With the continuous exploration of attack methods, Several digital-physical modeling techniques are widely recognized and used to improve the performance of physical attacks. Physical adversarial attacks on visible light have found that direct printing of adversarial perturbations can distort due to the printer. Non-Printability Score (NPS) (Shapira et al., 2023) has been devised to measure the distance between the adversarial perturbation and the printer. Furthermore, the natural world also suffers from noise and deformation problems. Expectation Over Transformation (EOT) (Athalye et al., 2018) has been proposed to consider potential transformations in the physical world. EOT discards the paradigm form to constrain the solution space. Instead, it utilizes the imposed expected distance between the adversarial inputs and the original inputs. It has been shown that it is difficult for cameras to capture extreme differences in neighboring pixels due to sampling noise (Sharif et al., 2016). To fit natural images' smoothness, the Total variation norm (TV) (Singh et al., 2022) is able to maintain perturbation smoothness.

## 2.3 MOTIVATION

As was previously indicated, several approaches have explored infrared physical adversarial attacks. However, all of these approaches generate adversarial patch by changing temperature of object surface, which is limited to deploy in physical world. In order to expand applicable scenarios, this paper builds physical infrared adversarial patch based on manipulating the infrared emissivity of object's surface.

Current physical adversarial patches are disposable while attacking different DNNs. In this study, a new material, galvanized iron sheets with different roughness, is used to generate infrared adversarial patch, and soft magnets are used as the background. Changing the arrangement order of galvanized patches with different roughness could generate a new infrared physical adversarial patch.

## 3 METHODOLOGY

This section presents our method. We first introduce the problem definition in Section A. Second, we explain the design of E-QR patch in Section B. Finally, we give the patch modeling and optimization methods of our attack in Section C.

### 3.1 PROBLEM DEFINITION

For an infrared detector $f(\cdot)$, the input is represented by an infrared image $x$. The output prediction, denoted as $f(x; \theta)$, is obtained by feeding the image $x$, where $x \in \mathbb{R}^{h \times w}$, to the detector $f$ with training parameter $P$. The attack image is crafted from adversarial patches, which is shown as:

$$x_{\text{adv}} = x \odot (1 - M) + P \odot M, \tag{1}$$

where $\odot$ signifies the Hadamard product, $M \in \{0, 1\}^{h \times w}$ denotes the mask matrix, and $P$ is the adversarial patch. The position and shape of the infrared patch depend on the matrix $M$. $M_{ij} = 1$ means that the position $(i, j)$ has a galvanized iron sheet. The degree of roughness of the galvanized iron sheet is determined by $P$.

The output $Y$ from the network $f$ comprises the position of the prediction box $Y_{\text{pos}}$ and the confidence level for the predicted class $Y_{\text{obj}}$. Our objective is to minimize the confidence score $Y_{\text{obj}} = f_{\text{obj}}$ for the object class in the network's predictions:

$$\arg\min Y_{\text{obj}} = \arg\min_{\delta} f_{\text{obj}}(x_{\text{adv}}). \tag{2}$$

This above optimization problem aims to find the adversarial patch $\delta$ that minimizes the confidence score for the object class, leading to potential vulnerabilities in the infrared detection system.

### 3.2 DESIGN OF E-QR PATCH

The mechanism of infrared imaging differs significantly from visible light imaging. Infrared images are gray-scale, where pixel values reflect the temperature of the object's surface. Larger pixel values

Figure 2: Design process of the E-QR patch. The purple region identifies the optimization process for the adversarial example. The orange region represents the process of making physical patch, using magnetism to attach galvanized iron sheet patches to a soft magnetic sheet arranged as physical IR counter patches. The blue region indicates the relationship between surface roughness and infrared emissivity, and shows the final physical infrared patches.

indicate higher infrared radiation intensity at the corresponding location. The infrared radiation intensity emitted from an object's surface is influenced by two factors, the infrared emissivity ($\mu$) and the temperature ($T$):

$$E = \varepsilon \mu T^4, \tag{3}$$

where $E$ is the radiance intensity of the materials surface, $\varepsilon$ is the Stephen Boltzmann constant, $\mu$ is emissivity of thin film and $T$ is the absolute temperature of the object. Depending on different radiation source, the radiation signal is calculated by the corresponding formula. The infrared emissivity of common materials is highly sensitive to the surface roughness of objects:

$$\varepsilon_r = [1 + (\frac{1}{\varepsilon_s} - 1)R]^{-1}. \tag{4}$$

In Eq.(4), $R$ is the emissivity of the roughness factor and represents the ratio of true surface area to apparent surface area. $\varepsilon_r$ and $\varepsilon_s$ express the emissivity of material's apparent surface and true surface.

Based on the mentioned relationship, patch patterns are designed using the material's roughness, as illustrated in Figure 2, resembling quick response (QR) codes in infrared images. In this paper, galvanized iron sheets are selected as the base materials. Besides, various polishing tools available in the manufacturing process can be used to achieve different roughness levels. As shown in Figure 2, white pixels represent that no modification is involved, reflecting the average temperature of the human body's surface. In contrast, black and gray pixels indicate that the infrared pixels are manipulated by galvanized iron sheets with different degrees of roughness. Figure 1 shows the pixel values corresponding to galvanized iron sheets with different degrees of roughness captured through an infrared camera. Consequently, it transforms the patch pattern design into a search optimization problem, exploring the positions of the candidate materials.

## 3.3 PATCH MODELING AND OPTIMIZATION

The digital patch $\boldsymbol{P}_d$ is a QR-like matrix ($N \times N$) generated in the digital domain. In order to make $\boldsymbol{P}_d$ realize in the physical domain, the TV norm is introduced to enable the aggregation of similar

---

**Algorithm 1** E-QR Patch Optimization

---

**Input:** Clean image $x$, Detector $f$, population size $Q$, the max number of iterations $t$
**Parameter:** A vector of parameter set $\boldsymbol{S}$
**Output:** Adversarial Image $x_{adv}$

1: Initialization: Randomly set $\boldsymbol{S}$.
2: **for** $k = 0$ to $t$ **do**
3:     Generate $\boldsymbol{S}^{k+1}$ based on crossover and mutation.
4:     **for** $i = 1$ to $Q$ **do**
5:         $P_d \leftarrow$ reshape $\boldsymbol{S}_i^k$.
6:         $P_p \leftarrow \mathbb{E}_{t \sim T}(P_d)$.
7:         $\boldsymbol{x}_{adv} \leftarrow P_p$ according to Eq.(1).
8:         $L \leftarrow L_{obj}(\boldsymbol{x}_{adv}), L_{TV}(P_d)$
9:         $\boldsymbol{S}^{k+1} \leftarrow$ the smaller one in $\boldsymbol{S}^{k+1}$ and $\boldsymbol{S}^k$ according to Eq.(5).
10:        **if** $f(\boldsymbol{x}_{adv})$ is NULL **then**
11:            **return** $\boldsymbol{x}_{adv}$
12:        **end if**
13:     **end for**
14: **end for**
15: **return** $\boldsymbol{x}_{adv}$

---

materials. Considering the distinction between the digital and physical realms, the transformation from digital to physical is simulated using the EOT method to obtain $\boldsymbol{P}_p$. $\boldsymbol{P}_p$ is then applied to objects in datasets, and the resulting images are fed into the object detector. To enhance the acquisition of physically adversarial and implementable patches, the loss is defined as:

$$L = L_{\text{obj}} + \lambda L_{\text{TV}}, \tag{5}$$

where $\lambda > 0$ serves as a small weight in the optimization algorithm, controlling and optimizing the shape of the patch to ensure the success of the attack.

$L_{\text{obj}}$ represents the loss calculated from the confidence of the detector in predicting the output of the patched image. EOT is a broad framework for enhancing the robustness of adversarial patching by considering a given transformation distribution $T$ during the optimization process, which can be defined as:

$$\tilde{\boldsymbol{P}} = \mathbb{E}_{t \sim T} \left( d(t(\boldsymbol{P}), t(\boldsymbol{P}')) \right), \tag{6}$$

where $\mathbb{E}_{t \sim T}$ denotes the EOT transform, $t(\cdot)$ is a transformation function chosen from the distribution $T$, including rotation, scale, noise, and so on. This constrains the expected effective distance between the adversarial outputs and the original inputs given the distance function $d(\cdot, \cdot)$. To enable the patch to deceive real-world object detectors, attempt a universal attack across different pedestrians. It is assumed that the attack dataset has $m$ images. The highest object prediction confidence score is selected as the score $Y_{\text{obj}}^i$ for each image $x_{\text{adv}}^i$. Then we have:

$$L_{\text{obj}} = \frac{1}{m} \sum_{i=1}^{m} \max(f_{\text{obj}}\left(\boldsymbol{x}_{\text{adv}}^i, \theta\right)). \tag{7}$$

$L_{\text{TV}}$ is designed to encourage the aggregation of the same material as much as possible within the patch. When the object is located at a considerable distance from the infrared sensor, fine details may be lost, diminishing the effectiveness of the attack. Simultaneously, placing the same material in close proximity facilitates the fabrication of adversarial patches. For a patch $\boldsymbol{\delta}$, we have

$$L_{\text{TV}}(\boldsymbol{P}) = \sum_{i,j} [(\boldsymbol{P}_{i,j} - \boldsymbol{P}_{i+1,j})^2 + (\boldsymbol{P}_{i,j} - \boldsymbol{P}_{i,j+1})^2]^{\frac{1}{2}}. \tag{8}$$

Considering that the selection of candidate materials is discrete and the attack on the detector is black-box, a nature-inspired optimization algorithm is considered to search the optimal results. In

this paper, Differential Evolution (DE) algorithm (Qin et al., 2008) is selected as the fundamental optimization tool, which consists of four components: initializing a population, generating offspring through crossover and mutation, selecting individuals with high fitness to survive, and preserving the best solution as the final result.

In our optimization, the patch is build as a individual $I$ of population. The population size is $Q$. The number of coding is matching the adversarial patch size that the patch has $N^2$ block while the $I$ has $N^2$ coding. The individual $I$ in population can be expressed:

$$I = \left\{ I_i^k | I_{ij}^k \in [0, n], 1 \le i \le Q, 1 \le j \le N^2 \right\}, \tag{9}$$

where $I_i^k$ is the $i$-th encoding which can reshape into QR-like patch, $k$ means the iterative number of generations, $I_{ij}^k$ represents the selection of which roughness for the block at position $j$ in the $i$-th individual. $n$ is the feasible domain of the decision space, representing the number of available emissivity to choose. The each block in have $n + 1$ state $I_{ij}^k \in [0, n]$, 0 means no sheet put in this location and other $n$ state means different roughness of the galvanized iron sheets. From this coding wo could determine the position and degree of roughness of the galvanized iron sheets. Various polishing tools available in the manufacturing process can be used to achieve different roughness levels. We make the prediction minimize of object influenced by our patch with EOT transform in digital domain to ensure it could affect in real world

The algorithm of generating the proposed E-QR patch is shown in **Algorithm 1**. The initial solution, $S^0$, is generated based on random initialization. The crossover and mutation among $S^k$ generates a new solution $S^{k+1}$. Evaluate the individuals $S_i^k$ in $S^k$ and $S^{k+1}$ using the fitness function 5 and select $Q$ individuals with the best fitness to form $S^{k+1}$ into a new round of optimization.

## 4 EXPERIMENTS

In this section, benchmark-based experiments are conducted to evaluate the efficiency of the proposed method. All experiments are performed on a windows server with 13th Gen Intel(R) Core(TM) i9-13900H CPU@2.60-GHz processor and a GPU server with 24G NVIDIA GTX 4090 GPU.

### 4.1 SIMULATION OF PHYSICAL ATTACKS

**Datasets:** We use the infrared images in the Teledyne FLIR ADAS Thermal dataset[1] to simulate the physical attacks. Following the (Wei et al., 2023a), We filter the original dataset for better fitting to the patch-based adversarial attack with three conditions. First, the images contains "person" category. Second, the height of the object bounding box of the person in images exceeds the 120 pixel value. Third, human bodies have no overlap in the images. Finally, the attack dataset include 378 available images with 479 eligible "person" labels. The target detector's AP was $100\%$ for the clean images in attack dataset.

**Target Detector:** For pedestrian detection task, we choose You Only Look Once (YOLO) target detector of YOLOv5[2] because of its fast speed. We used the pretrained weights on the MSCOCO Dataset (Lin et al., 2014) and fine tuning on the FLIR ADAS datasets. This model is used as the target model in our attack process. These models are then used as the target models in our attack process.

**Attack Methods:** We compare the proposed method with three state-of-the-arts attack methods, i.e., Infrared Invisible Clothing (Zhu et al., 2022), Hotcold Block (Wei et al., 2023a) and Irregular Patch (Wei et al., 2023b). The Infrared Invisible Clothing and Irregular Patches are gradient-based attack methods, and Hotcold Block is based by genetic algorithm.

**Parameter Setting:** In the DE algorithm, we set the number of the initial population as 50, and the epochs of evolution as 100. The smooth galvanized iron sheet pixel value is set 0.1, the rough galvanized iron sheet pixel value is 0.3, and the soft magnetic sheet pixel value is 0.5.

---

[1]https://www.flir.com/oem/adas/adas-dataset-form/
[2]Jocher, G. 2020. https://github.com/ultralytics/yolov5

Table 2: Quantitative results for the attack dataset in different settings. We report the AP (%), ASR (%) with our adversarial attack method, E-QR patch (E-QR), versus the Random QR code patch in YOLOv5 detector, for different patch pixel bit depth value and resolution of patch (side number).

| Pixel Bit Depth | Method | Side Number | | | | | | | | | | | | |
|---|---|---|---|---|---|---|---|---|---|---|---|---|---|---|
| | | 5 | | 6 | | 7 | | 8 | | 9 | | 10 | | Average | |
| | | AP(%) | ASR(%) | AP(%) | ASR(%) | AP(%) | ASR(%) | AP(%) | ASR(%) | AP(%) | ASR(%) | AP(%) | ASR(%) | AP(%) | ASR(%) |
| 2 | Random | 53.5 | 62.3 | 58.8 | 58.7 | 58.3 | 60.1 | 65.2 | 52.2 | 68.2 | 45.7 | 66.9 | 50.7 | 61.8 | 54.9 |
| | E-QR | 41.3 | 86.2 | 42.8 | 78.9 | 40.8 | 78.2 | 45.3 | 76.1 | 49.2 | 71.7 | 47.7 | 72.4 | **44.5** | **77.3** |
| 3 | Random | 58.8 | 54.3 | 58.1 | 54.3 | 59.9 | 50.7 | 66.1 | 44.2 | 69.7 | 41.3 | 69.1 | 42.2 | 63.6 | 47.8 |
| | E-QR | 41.6 | 85.4 | 43.8 | 75.6 | 45.8 | 67.5 | 46.0 | 76.1 | 49.4 | 71.0 | 51.8 | 66.7 | **46.4** | **73.7** |
| 4 | Random | 59.4 | 49.3 | 61.2 | 52.9 | 61.7 | 47.1 | 68.8 | 39.1 | 68.7 | 41.3 | 69.1 | 39.8 | 64.8 | 45.3 |
| | E-QR | 42.4 | 76.1 | 43.4 | 76.1 | 45.1 | 73.2 | 50.0 | 70.2 | 53.4 | 65.9 | 54.0 | 65.2 | **48.1** | **71.1** |
| 5 | Random | 59.1 | 50.0 | 59.2 | 50.1 | 60.9 | 49.6 | 68.3 | 41.3 | 70.6 | 39.9 | 67.7 | 41.3 | 64.3 | 45.4 |
| | E-QR | 43.5 | 79.7 | 44.6 | 75.4 | 48.4 | 71.7 | 50.0 | 70.2 | 53.8 | 63.8 | 57.4 | 63.0 | **49.6** | **70.6** |
| 6 | Random | 60.4 | 50.7 | 60.0 | 49.3 | 61.3 | 44.9 | 66.7 | 41.3 | 67.9 | 37.0 | 68.9 | 40.5 | 64.2 | 43.9 |
| | E-QR | 42.7 | 82.2 | 42.9 | 78.3 | 45.8 | 78.2 | 50.0 | 70.2 | 55.3 | 66.7 | 56.5 | 61.6 | **48.9** | **72.6** |
| 7 | Random | 58.9 | 51.4 | 60.3 | 49.3 | 61.9 | 44.9 | 68.1 | 42.8 | 68.0 | 37.6 | 68.4 | 42.0 | 64.3 | 44.7 |
| | E-QR | 40.1 | 85.9 | 41.8 | 84.8 | 46.6 | 77.5 | 47.9 | 73.2 | 52.6 | 69.6 | 55.9 | 62.3 | **47.5** | **75.6** |
| 8 | Random | 60.9 | 49.3 | 59.8 | 49.3 | 61.9 | 44.9 | 68.7 | 39.1 | 68.9 | 38.4 | 68.5 | 41.3 | 64.8 | 43.7 |
| | E-QR | 42.7 | 80.4 | 42.5 | 81.9 | 43.8 | 79.0 | 49.0 | 73.9 | 50.5 | 70.2 | 56.2 | 61.6 | **47.5** | **74.5** |
| 9 | Random | 58.7 | 50.0 | 62.0 | 47.1 | 60.1 | 45.7 | 68.8 | 39.1 | 68.5 | 37.0 | 68.1 | 42.8 | 64.4 | 43.6 |
| | E-QR | 41.9 | 83.3 | 41.7 | 83.3 | 43.4 | 79.7 | 46.4 | 74.6 | 51.0 | 69.6 | 55.8 | 62.6 | **46.7** | **75.5** |
| 10 | Random | 58.7 | 50.0 | 62.0 | 48.3 | 59.6 | 46.9 | 68.7 | 39.1 | 67.9 | 36.2 | 69.4 | 40.9 | 64.4 | 43.5 |
| | E-QR | 40.8 | 88.4 | 41.3 | 86.2 | 43.2 | 81.2 | 47.4 | 73.2 | 51.8 | 80.4 | 55.5 | 63.8 | **46.5** | **78.8** |

**Performance Metrics:** Attack Success Rate (ASR) and Average Precision (AP) are used to evaluate the attack performance. ASR denotes the ratio of successfully attacked images out of all the test images. AP is computed by measuring the region under the Precision-Recall (PR) curve.

### 4.1.1 QUANTITATIVE EXPERIMENT

The quantitative experiment is to attack each image in the attack dataset with different settings for patch pixel bit depth and patch resolution. The effect of patch resolution and the bit depth of each pixel on the attack effect is explored, where patch resolution quantitative analysis is done by varying the total number of patch cells with the same area. Pixel bit depth quantitative analysis is to change the number of pixel values that can be selected in each patch cell, e.g., when the bit depth is 2, the pixel values that can be chosen are 0 and 0.5. When the bit depth is 3, the pixel values that can be selected are 0, 0.25, 0.5, and so on after that. Table 2 reports the results of evaluating the attack effectiveness of our method (E-QR) with random patches. According to the above results, we can draw the following conclusions. The proposed E-QR patch outperforms random patches across the board. The effectiveness of the attack decreases as the resolution of the patches gradually increases, which we attribute to the sharp rise in the number of patches to be optimized, leading to an increase in the difficulty of solving the parameters and making it difficult to optimize a higher-quality solution in the same amount of time. Under the influence of the patch pixel bit depth, the attack effect at the ends of the parameter interval performs better than the middle. Therefore, choosing the $5 \times 5$ resolution for the E-QR patch is reasonable. Subsequent experimental alignments are conducted according to this configuration unless otherwise specified.

### 4.1.2 COMPARISON WITH SOTA ATTACKS

We attacked each image in the validation set of the FLIR ADAS dataset and compared it with other methods. In Figure 3, we plotted the Precision-Recall (P-R) curve for evaluating YOLOv5 and show qualitative examples of various baseline methods. In the P-R curve, our approach demonstrates robust competitive performance. The E-QR patch resulted in a 52.3% decrease in the AP of the YOLOv5 detector and 81.5% ASR, significantly outperforming the 10.6% drop from the Irregular Patch and the

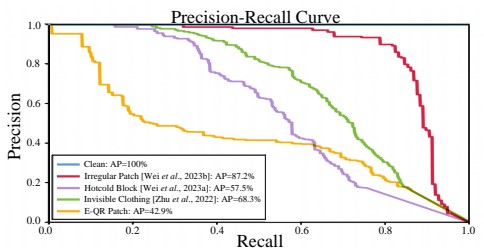

Figure 3: Precision-Recall Curve of E-QR Patch and SOTA Methods

26.2% drop from Infrared Invisible Clothing, and marginally surpassing the 37% drop from the Hot-cold Block.

## 4.2 Physical Attacks

We tested the performance of E-QR Patch in the physical world, and the physical experimental environment setup is shown in Appendix A. We record 5 videos in various settings using frame extraction per second in our physical experiment. In total, 243 images were captured, encompassing 316 pedestrian labels. These images are trained by YOLOv5, with a detector confidence threshold set at 0.7.

### 4.2.1 Attacks in Different Scenarios

We investigated the robustness of the E-QR patch in two key aspects: distance and human posture. The experiment result details are shown at Appendix B. Initially, we tested the attack success rate (ASR) as a person, holding the adversarial patch, moved away from an infrared detector, starting at 1m. The average ASR was found to be 81.9%, with specific rates of 93.8% at distances of 1-2m, 80.5% at 3m, and a significant drop to 57.6% at distances greater than 4m, attributed to the patch's gradual deformation as distance increased. Subsequently, we assessed the impact of human posture and environmental factors by having a person stand 2m away from the detector while performing various movements such as angular rotations, standing up, and sitting down. E-QR Patch perform well in small rotations, both standing and sitting postures exhibited good robustness. However, with big rotations, crucial information from the patch became obscured, leading to a sharp decline in ASR. Furthermore, the effectiveness of the E-QR Patch is affected by the infrared radiation of different environments in different temperature fields.

## 4.3 Evaluation of Robustness

We evaluated the robustness of our method in a black-box setting against Faster RCNN (Ren et al., 2015), Mask RCNN (He et al., 2017) YOLOv3 (Redmon & Farhadi, 2018), YOLOv8 detector[3]. These detectors were pretrained on the MSCOCO dataset and fine-tuning on the FLIR ADAS dataset. Table 3 reports the

| Detector | Clean AP (%) | Attack AP (%) | ASR (%) |
|---|---|---|---|
| Faster RCNN | 94.6 | 16.4 | 93.2 |
| Mask RCNN | 97.4 | 18.7 | 92.7 |
| YOLOv3 | 96.6 | 19.5 | 91.5 |
| YOLOv5 | 99.2 | 42.9 | 81.5 |
| YOLOv8 | 95.7 | 59.0 | 54.4 |

Table 3: Evaluation across various detectors.

changes in ASR and AP. It is evident that the performance of other DNN detector significantly decreases when subjected to the E-QR patch attack. The EQR patch was effective on networks published before YOLO v5. It was not effective on the latest network, YOLO v8. We consider that YOLO v8 uses more residual units and a Decoupled Head structure in model training that improves the robustness of the model.

## 5 Conclusion and Discussion

This paper introduces an adversarial attack for infrared detectors, known as the E-QR patch. This innovative approach involves utilizing the different roughness of galvanized iron sheets to modify the infrared emissivity of object surfaces, creating adversarial patches characterized by intricate textures. Consequently, these patches hide individuals by fooling the detector with infrared sensors. Moreover, we have developed a reusable physical adversarial carrier by exploiting the magnetic properties of soft magnetic sheets to adhere galvanized iron sheets. This innovative carrier system enhances the practicality and sustainability of the adversarial attack, contributing to its real-world applicability. A comprehensive set of experiments conducted in both digital and physical domains provides compelling evidence for the effectiveness of our E-QR patch in successfully circumventing detection models. In the future, we hope to overcome the shortcomings of physical objects at different scales, putting the adversarial patch on 3D-based modeling to improve the robustness of scales. Moreover, emissivity-based adversarial attack patches can be combined with infrared stealth technology to achieve cross-band adversarial attacks.

---

[3]Jocher, G. 2023. `https://docs.ultralytics.com/`

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

|  (a)1m  |  (b)2m  |  (c)3m  |  (d)4m  |

Figure 4: Example results of digital attacks. The bounding boxes indicate the infrared detector successfully detects the person.

## A  PHYSICAL REALIZATION

Galvanized iron sheets are used as the primary material for fabricating the patches. Varying degrees of sanding were applied to modify the emissivity of their surfaces, as shown in Figure 1. The patches consist of a smooth sheet of galvanized iron to achieve a low reflectivity surface and a rough sheet of galvanized iron to enhance its surface emissivity. These two different roughness levels were used to finalize the physical patch by affixing the iron sheet to a soft magnetic sheet with magnetic property. The galvanised iron sheet is easy to paste or remove on the soft magnetic sheet, so a soft magnetic sheet can be multiplexed with a variety of physical adversarial patch patterns. The thermal emissivity of the soft magnetic sheet is similar to conventional clothing so adhered to the human body over an extended. Patches of identical dimensions were also crafted from all three materials to validate the efficacy of our patches. The attack performance of these clothes was then tested in the real world.

We employed the FLIR ONE Pro camera with a thermal resolution of $160 \times 120$ for infrared imaging. Throughout the capture process, the camera is connected to a Xiaomi phone, enabling real-time image display. We captured images of individuals in various indoor and outdoor scenarios, with the distance between the camera and the subjects ranging from 1 to 4 meters. The images depict individuals in different poses, such as standing and sitting.

## B  PHYSICAL EXPERIMENT

We first explored the robustness of the E-QR patch over distance by testing the attack success rate with the detected person looking squarely at the infrared detector, holding the antagonistic patch, and starting at a distance of 1m and gradually moving backward by 1m. The average ASR is $81.9\%$. As shown in Figure. 4, E-QR patch achieves an $93.8\%$ ASR at a distance including 1-2 meters, $80.5\%$ ASR in the distance of 3m and $57.6\%$ ASR with the distance more than 4m. Due to the resolution of the detector, the adversarial patch gradually deforms as the distance of the detected person increases, resulting in a rapid decrease in ASR.

We then explored the robustness of the E-QR patch to human posture as well as the environment. The detected person stood 2 meter away from the infrared detector holding the counter patch, and performed angular rotation as well as standing up, sitting down, etc., as shown in Figure. 5. The human body posture and the environment were then investigated. For small rotations, both standing and sitting postures show good robustness with an ASR of $92.6\%$, while after the angle exceeds $30°$, part of the patch's information is obscured and lost, and the ASR drops sharply. In the outdoor environment, where the ambient background temperature is 0-5°C and subject to a lot of infrared interference in the environment, the gap between the display effect of patch and that of indoors becomes larger, and the average ASR drops to $82.4\%$.

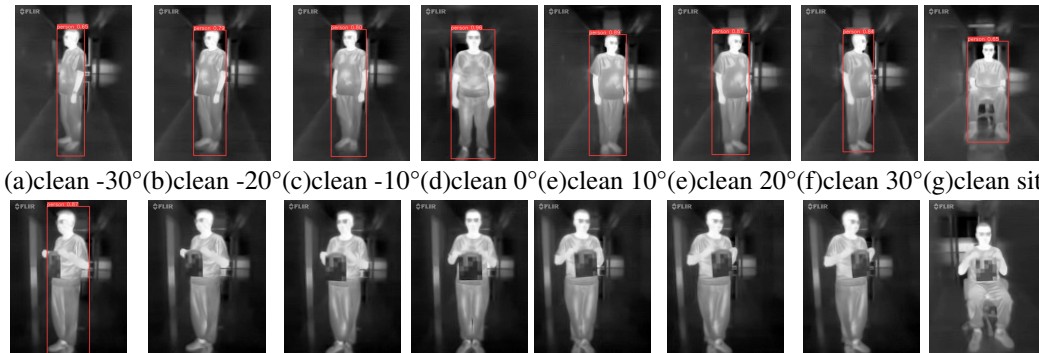

(a)clean -30°(b)clean -20°(c)clean -10°(d)clean 0°(e)clean 10°(e)clean 20°(f)clean 30°(g)clean sit

(h)patch -30°(i)patch -20°(j)patch -10°(k)patch 0°(l)patch 10°(m)patch 20°(n)patch 30°(o)patch sit

Figure 5: Visual examples of physical attacks with infrared patches under various angles, postures. The bounding boxes indicate the infrared detector successfully detects the person.

## C    EFFECT OF $\lambda$

$\lambda$ is a parameter that balances the adversarial loss as well as the Total variation norm, and a larger $\lambda$ will make the patch more inclined to the less varied patches.We investigated values of $\lambda$ with lambda of 0, 0.01, 0.05, 0.1, and 0.2. As $\lambda$ gets larger, the value of $L_{TV}$ gets smaller, but the success rate of the attack gets lower. The more complex the variation of the patch the more effective the attack will be.

## D    DEFENSE DISCUSSION

We discussed defense strategies against the E-QR patch. Adversarial training was employed to enhance the model's robustness (Goodfellow et al., 2014b). Specifically, adversarial examples generated by the E-QR patch were added to the training set, and the original model was retrained. Subsequently, the newly trained network was attacked again using the E-QR patch to assess its effectiveness in the digital space. Comparing the results, the retrained network achieved an AP of 96.6% without attack and 90.0% after the attack. The retrained model exhibited increased robustness with only marginal performance loss. Therefore, our image enhancement approach can further improve the detector's performance. In practical applications, this holds significant implications for deploying deep learning models in real-world scenarios.

