# OpenReview forum: "Manipulating Infrared Emissivity with Galvanized Iron Sheets for Physical Adversarial Attack"
_ICLR.cc/2025/Conference — ICLR 2025 Conference Withdrawn Submission_

### Official Review · Reviewer_SJNX · 2024-10-17

**Soundness:** 2
**Presentation:** 2
**Contribution:** 1
**Rating:** 3
**Confidence:** 5

**Summary:**

Overall, the paper lacks innovation. Although it proposes to use the emissivity change of galvanized iron sheets to achieve infrared adversarial attacks, it does not make good use of this feature and still uses limited pixel blocks to express adversarial maps. At the same time, the paper also uses existing methods in optimization methods and does not design a more suitable optimization algorithm for infrared scenes.

**Strengths:**

The paper realizes infrared physical adversarial patches through the design of galvanized iron sheets, which makes the color of countermeasure mapping in the infrared world more controllable.

**Weaknesses:**

1. Using materials with different emissivity to implement infrared adversarial attacks is not novel enough. In the paper "Multispectral Invisible Coating: Laminated Visible-Thermal Physical Attack against Multispectral Object Detectors using Transparent Low-e films", the author used materials with different emissivity to implement multispectral adversarial clothing, which is obviously more solid and applicable. In "Infrared Adversarial Car Stickers", the author also used the difference in emissivity to implement infrared adversarial attacks.
2. The author did not innovate in the design of the optimization algorithm and the method of generating color blocks, and all used the previous method.

**Questions:**

1. Since the color of the infrared map can be controlled by galvanizing, why is it still optimized to a grid form? Can it be optimized to a more complex pattern (more colors and pixels)?
2. Is it possible to simultaneously achieve adversarial attacks in the visible light world by controlling the color of the material?
3. The author mentioned the TV loss, but as a grid map with clear boundaries and a relatively small number of pixels, does this loss make sense?
4. In addition to EOT, do the authors use other methods to bridge the gap between the digital and physical worlds? Because the number of optimized patch colors is limited, using color mapping seems to be more effective.

---

### Official Review · Reviewer_HPgv · 2024-11-03

**Soundness:** 3
**Presentation:** 2
**Contribution:** 2
**Rating:** 5
**Confidence:** 5

**Summary:**

This paper analyzes the possibility that current physical attack techniques mainly perform physical attacks by manipulating the temperature profile of the target surface, while ignoring the physical attack caused by the physical property of infrared emissivity.  The authors perform an attack on the IR detector by manipulating the roughness of the object surface to change the surface emissivity of the object.  It has been verified that the method is effective in digital environment and physical environment.

**Strengths:**

The authors explored and found that the use of metals with different roughness can perform infrared attacks.

**Weaknesses:**

Here are some of my concerns:
(1) In view of the fact that the patch deployed on the clothing will cause the clothing to sag, what is the quality of the physical patch generated by the author?
(2) Irregular reference in line 38: This paper [1] is the first paper on adversarial attacks, and it is unreasonable to quote here.
(3) The author has not fully investigated the literature, and there is a lack of more published literature on infrared physical attack in Related works, such as literature [2-4] and so on.
(4) The author mentioned "and Hotcold Block is based by genetic algorithm." in line 372 of the paper, however, HCB[5] used PSO to optimize, not "genetic algorithm".
(5) For infrared physical attack, in view of the difficulty of physical implementation, the mainstream practice uses a single color (black) or two colors (black and white) perturbation to perform optimization and attack.  In the author's physical attack, the color presented by the perturbation is variable, but even so, it is a limited number of visual presentation effects.  Therefore, the optimization variables involved in the simulation optimization in physical attacks will not be too complex.
(6) The author sets the detection threshold of the target detector to 0.7, which will lead to an increase in the success rate of the attack.  Generally, the threshold is set to 0.5.  Authors are advised to perform experiments with the threshold set to 0.5.

[1] Christian Szegedy, Wojciech Zaremba, Ilya Sutskever, Joan Bruna, Dumitru Erhan, Ian Goodfellow, and Rob Fergus. Intriguing properties of neural networks. arXiv preprint arXiv:1312.6199, 2013.
[2] Zhu X, Liu Y, Hu Z, et al. Infrared Adversarial Car Stickers[C]//Proceedings of the IEEE/CVF Conference on Computer Vision and Pattern Recognition. 2024: 24284-24293.
[3] Zhu X, Hu Z, Huang S, et al. Hiding from infrared detectors in real world with adversarial clothes[J]. Applied Intelligence, 2023, 1-19.
[4] Hu C, Shi W, Yao W, et al. Adversarial Infrared Curves: An attack on infrared pedestrian detectors in the physical world[J]. Neural Networks, 2024: 106459.
[5] Wei H, Wang Z, Jia X, et al. Hotcold block: Fooling thermal infrared detectors with a novel wearable design[C]. Proceedings of the AAAI conference on artificial intelligence, 2023: 15233-15241.

**Questions:**

See Weakness

---

### Official Review · Reviewer_G5Qf · 2024-11-03

**Soundness:** 3
**Presentation:** 3
**Contribution:** 3
**Rating:** 6
**Confidence:** 5

**Summary:**

The paper proposes an adversarial attack method targeting pedestrian detectors in the infrared domain by exploiting the emissivity properties of galvanized iron sheets, a feature previously overlooked, as past research has centered on temperature variations. Specifically, the proposed method crafts adversarial patches resembling a QR-code pattern, where the pixel intensity is modulated by adjusting the roundness of the galvanized iron sheet’s surface. The optimization leverages a differential evolution method, constrained by Expectation over Transformation (EOT) and Total Variation (TV) norms. Experimental results demonstrate the method’s efficacy under certain conditions, outperforming baseline methods in key settings.

**Strengths:**

1. The creative approach of utilizing galvanized iron sheets, whose emissivity varies based on surface roughness, adds a novel angle to the adversarial attack landscape.

2. The experimental design is thorough, incorporating ablation studies on patch pixel depth and resolution, testing across varied distances and angles, and evaluating on black-box detectors.

3. The proposed attack shows strong generalizability across different detectors, such as Faster RCNN, Mask RCNN, and YOLOv3, although it would benefit from further adaptation to stay competitive with the latest YOLO versions.

**Weaknesses:**

1. The paper omits a relevant reference [*] that uses reflective and insulation plastic tapes to manipulate intensity distribution in near-infrared images. Unlike the current method, this approach applies tapes over the entire body to achieve pose and direction invariance, as well as increased robustness at longer distances. The authors should discuss how their approach compares to or differs from the tape-based method, particularly regarding pose/direction invariance and robustness at longer distances.

2. As the patch is affixed to a single side, the attack faces a trade-off between direction invariance and stealth. Consequently, the attack’s effectiveness is constrained to a ±30° angle. It would be beneficial for the authors to discuss potential solutions or future work to extend the effective angle range while maintaining stealth.

3. With galvanized iron sheet patches, achieving adversarial effects by controlling surface roughness can be labor-intensive, posing practical challenges in producing precise patches. The authors could enhance the discussion by suggesting possible solutions to make this process more efficient or precise.

4. The paper would benefit from an ethics discussion, including a section addressing both the positive implications (e.g., improving detector robustness) and the risks (e.g., misuse to evade detection), along with any proposed mitigation strategies.

Reference:
[*] Niu, Muyao, Zhuoxiao Li, Yifan Zhan, Huy H. Nguyen, Isao Echizen, and Yinqiang Zheng. "Physics-Based Adversarial Attack on Near-Infrared Human Detector for Nighttime Surveillance Camera Systems." In Proceedings of the 31st ACM International Conference on Multimedia, pp. 8799-8807. 2023.

**Questions:**

Please address the concerns outlined in the weaknesses section.

---

### Official Review · Reviewer_EQJr · 2024-11-04

**Soundness:** 2
**Presentation:** 3
**Contribution:** 2
**Rating:** 5
**Confidence:** 5

**Summary:**

This paper proposes the E-QR patch, using galvanized iron sheets as an adversarial medium to attack infrared person detectors. The authors discovered that certain materials can alter infrared emissivity, thereby affecting infrared imaging. The E-QR patch optimizes the position and degree of roughness of the galvanized iron sheets using a DE algorithm to achieve the attack. The authors verified the attack performance of the E-QR patch in both digital and physical spaces.

**Strengths:**

This paper explores the relationship between infrared emissivity and infrared imaging, and introduces a new material, galvanized iron sheets, as an adversarial medium, which is interesting.

**Weaknesses:**

1. The authors claim that all existing approaches generate adversarial patches by altering the temperature of the object surface, but Infrared Invisible Clothing [1] uses aerogel, which, to my knowledge, does not change temperature. Does this proposed method share design similarities with Infrared Invisible Clothing, including QR-like patterns?
2. Implementation details. Please provide details on fine-tuning YOLOv5, as different parameter settings can affect the detector’s robustness, which could introduce bias into subsequent attack performance evaluations.
3. Limited novelty. This paper mainly introduces a new material for implementing physical adversarial attacks. The algorithm and digital modeling of adversarial patches are quite similar to existing methods.
4. Lack of ablation study. The impact of EOT Transfer and the hyperparameters of the DE algorithm in the method framework on attack performance is unclear.

[1] Infrared Invisible Clothing: Hiding from Infrared Detectors at Multiple Angles in Real World. CVPR 2022.

**Questions:**

1. In the digital attack, what are the confidence and IoU thresholds? In the physical attack, what is the IoU threshold set for calculating AP?
2. The dataset used in this paper contains only 378 available images with 479 eligible "person" labels. How were the training and validation sets divided? Can such a small sample size adequately support fine-tuning YOLOv5, and is there a risk of overfitting?

---

### Note · Authors · 2024-11-13

I have read and agree with the venue's withdrawal policy on behalf of myself and my co-authors.